# Electrochemical Determination of Morin in Natural Food Using a Chitosan–Graphene Glassy Carbon Modified Electrode

**DOI:** 10.3390/s22207780

**Published:** 2022-10-13

**Authors:** Edgar Nagles, Monica Bello, John J. Hurtado

**Affiliations:** 1Facultad de Química e Ingeniería Química, Universidad Nacional Mayor de San Marcos, Lima 15081, Peru; 2Facultad de Ciencias Naturales y Matemáticas, Universidad Nacional Federico Villareal, Lima 15001, Peru; 3Departamento de Química, Universidad de Los Andes, Carrera 1 No. 18A-12, Bogota 111711, Colombia

**Keywords:** glassy carbon, chitosan, graphene, morin, square wave voltammetry

## Abstract

This report presents a new application for the chitosan–graphene glassy carbon electrode (Ch-G/GCE) system in the determination of the hydroxyflavonoid morin (MR), one of the flavonoids with the highest favorable activity for people, due to its natural properties by square-wave voltammetry (SWV). The anodic peak current for MR was observed at 0.50 V with an increase of 73% compared with the glassy carbon electrode unmodified. The surface areas of Ch-G/GCE, Ch/GCE and GCE evaluated by cyclic voltammetry were 0.140, 0.053 and 0.011 cm^2^, respectively. Additionally, an increase greater than 100% compared to the electrode without modification was observed. The detection limit was 0.30 µmol/L for MR, and the relative standard deviations (RSDs) were 1.8% (*n* = 6). Possible interferences as quercetin, rutin, and applications in real samples were also evaluated with very acceptable results.

## 1. Introduction

Morin, with IUPAC nomenclature 2′,3,4′,5,7-pentahydroxyflavone, is a plant-derived polyphenol compound with high antioxidant, antidiabetic, anti-inflammatory, antitumor, antihypertensive, and antibacterial activities, which can be consumed as a medicine [1]. This property increases interest in the development of methodologies that allow for the detection and quantification of foods, drugs, and natural products, especially when MR is used as a new natural medicine for effective cancer treatments [2]. Some reports for MR detection have used techniques such as fluorescence [3,4,5,6] and HPLC [7]. According to these reports, the detection limits are very sensitive, between 0.04 and 0.6 µmol/L, and are applied in biological samples and natural products. On the other hand, although they are very selective and sensitive methodologies, they have a high instrumental cost.

One of the best alternatives for developing low-cost but equally sensitive and selective alternative methodologies for MR detection may be electroanalytical techniques. These results present an innovation in the development of the working electrode. In the last decade, the number of reports on detecting MR using different types of modified working electrodes has increased; in some reports, carbon nanotubes [8,9], coordination complexes such as MoS_2_ nanosheets [10], Tb_2_Se_2_ nano-octagon integrated oxidized carbon nanofibers [11], Zn oxide [12], Ni(II) phthalocyanine [13], Hexamine cobalt(III) [14], and nanoparticles such as Au nanoparticles–graphene [15] and Ag nanoparticles–graphene 1 [16] have been used. According to these reports, the detection limits are very sensitive, between 0.60 and 0.002 µmol/L, and MR oxidation is observed at relatively low potentials, between 0.2 and 0.4 V vs. Ag/AgCl. MR oxidation involves equally sensitive methodologies compared to the techniques mentioned above. On the other hand, the use of cheaper but equally sensitive electrodes has been reported with the use of chitosan alone. Previous reports have used chitosan to modify carbon electrodes to detect thimerosal [17] and sunset yellow [18]. Moreover, electrochemical impedance studies of chitosan-modified electrodes for application in electrochemical sensors demonstrate that the properties of these substances enhance charge transfer [19].

The properties of graphene in electrode modification and capacitors are widely recognized. Currently, several reviews have reported on its applications, such as the performance of microbial fuel cells, from 2015 and 2016 [20,21]; the modification of electrodes, from 2012, 2013, and 2018 [22,23,24]; the development of biosensors and sensors, from 2010 and 2013 [25,26]; and sensors for dopamine, from 2009 [27]. The advantages of this emerging material are mainly related to the greater efficiency in charge transfer, an increase in the active surface area, a wide electrochemical window, and better catalytic activity in processes that are dominated by metal catalysts when it acts as an anode or a cathode [28].

On the other hand, graphene has been used in electrochemical devices, sensors, biosensors, energy generation, and storage [29]. In the development of electrochemical sensors, the increased use of graphene is due to the ease of functionalizing graphene surfaces with specific elements of interest [29]; in this case, chitosan with glassy carbon has the ability to form stable films on surfaces [30,31]. Additionally, the active sites of graphene can be improved with dopants such as boron nitride [32] and alanine [33]. Using computational chemistry calculations, it has been shown that active sites for electron transfer are enhanced on the surface of graphene [32,33].

The combined properties of chitosan and graphene on gold electrode modification were initially reported in the development of glucose sensors in early 2009 [34]. That same sensor, a year later, was reported with the addition of Au nanoparticles on the gold electrode [35]. Additionally, in that same year, it was reported for the simultaneous detection of dopamine, uric acid, and ascorbic acid on glassy carbon electrodes [36,37]. In addition, this system has been used for the detection of peroxide [38], nitrite [39], and rutin [40,41]; and metallic cations such as lead and cadmium [42]. In most of these reports, the chitosan–graphene system is combined with other substances, such as metal nanoparticles, polymers, metal oxides, and carbon nanotubes. On the other hand, the authors report low detection limits, and high selectivity and reproducibility. Furthermore, the detection of morin has not yet been reported. All of this evidence and the morin properties justify the development of this methodology and its application in the analysis of mass consumption samples such as coffee and tea. Moreover, the motivation for the development of new electroanalytical methodologies using electrodes modified with graphene and chitosan is based on the great variety of applications, with the main objective of developing a methodology of easy reproduction, application, and reliability that allows good results compared to commonly used techniques, but with a lower cost and a reduction in analysis time.

In the present report, a new methodology was developed that uses a simple chitosan–graphene glassy carbon electrode to detect morin in foods via square wave voltammetry between 0.0 and 1.0 V. Electrode characterization was performed via cyclic voltammetry, and the active area of the electrode was increased by more than 100% compared to the unmodified electrode. Statistical parameters as a detection limit were less than 1.0 µmol/L, and the linearity range was greater than 1.0 µmol/L. In addition, the matrix effect of the real samples was not significant.

## 2. Materials and Methods

### 2.1. Reagents and Instruments

All reactants, such as ultrapure water, phosphoric acid, NaH_2_PO_4_ and Na_2_HPO_4_ salts, morin, rutin, KCl, and K_4_Fe(CN)_6_, were obtained from Merck (Sigma–Aldrich, Darmstadt, Germany). Phosphate-buffered solution were prepared at pH levels of 2.0, 3.0, 4.0, 5.0, and 6.0, at a concentration of 10.0 mmol/L. An Autolab PGSTAT204 potentiostat/galvanostat (Metrohm, Herisau, Switzerland) was used in cyclic voltammetry (CV), and square wave voltammetry (SWV) measurements were made with a glassy carbon electrode (GCE 2 mm internal diameter from Chi-Instrumen, China), a glassy carbon electrode modified with chitosan film (Ch/GCFE), and a glassy carbon electrode modified with chitosan and graphene film (Ch-G/GCFE), as working electrodes. The reference and auxiliary electrodes used were Ag/AgCl with KCl (3.0 mol/L) and a platinum wire, respectively, from Chi-Instrument, China.

### 2.2. Electrode Preparation of Ch/GCE and Ch-G/GCE

Before each measurement, the glassy carbon electrode (GCE) surface was treated with alumina powder with a particle size of between 0.3 and 0.05 μm. Next, GCEs were placed in a beaker with 1.0 mol/L HNO_3_ that was then placed in an ultrasound bath for 2 min to obtain a completely clean surface. Subsequently, a chitosan solution was prepared only once for the entire study, using 5.0 mg of low molecular weight chitosan (Sigma–Aldrich) that was dispersed in 10.0 mL of acetic acid (1%). Freshly prepared chitosan solution (30.0 μL) was deposited on the surface of freshly cleaned glassy carbon heated at 80 °C. The surface was left for 5 min to remove excess solvent [18]. In this way, the first modified electrode, Ch/GCE, was manufactured. The modified Ch-G/GCE was prepared with a dispersion of 0.1 mL of graphene oxide (Sigma Aldrich) in 0.5 mL of freshly prepared chitosan solution. Thirty microliters were incorporated on the GCE surface, and the electrode was treated in the same way as that developed for the Ch/GCE. Only the freshly modified Ch-G/GCE electrode was treated, by applying a potential constant of −1.0 V for 60 s in phosphate-buffered solution with a pH of 7.0 to reduce the functional groups that the graphene oxide contains. This treatment must be carried out for the results to be reproducible. Next, Ch-G/GCE was optimized with three consecutive cycles of a potential scan between 0.0 and 1.2 V at 0.1 V/s with cyclic voltammetry to obtain a clean and homogeneous modified film surface [43]. The chitosan film formed does not dissolve in an acidic medium used in PBS.

### 2.3. Sample Preparation

Ground coffee (2.0 g) was mixed with 50 mL of hot water for 20 min, and then the mixture was filtered using filter paper. A tenth to a half mL of the cold extract was added to the electrochemical cell without any previous dilution, to perform the measurements. The aromatic chamomile beverage, whose mass was approximately 3.3 g, was mixed with 10 mL of hot water for 20 min, after which the extract was cooled before adding 0.01–0.1 mL to the cell for the development of measurements. All samples were obtained from a supermarket in Lima, Peru.

### 2.4. Measurement Procedure

CV was used for electrode surfaces characterized using the test electrolyte K_4_Fe(CN)_6_, with 5.0 mmol/L KCl and an MR activity of 0.12 mmol/L. Cyclic voltammograms were scanned from −0.3 to 1.0 V, at scan rates of between 0.02 and 0.12 V/s. SWV was used for quantification at 0.0 V for 30.0 s, with a frequency of 15.0 Hz and a pulse amplitude of 50.0 mV. The electrochemical reaction was completed with ultrapure water (9.0 mL), phosphate-buffered solution (0.3 mL), and 5.0 mmol/L MR (250.0 µL) using CV, and 0.50 mmol/L MR (20.0 µL) for all SWV studies.

## 3. Results and Discussion

### 3.1. GCE, Ch/GCE, and Ch-G/GCE Surfaces and MR Activity

The authors mainly focused on the study of the surface area modified electrode characterization using CV because the surfaces of similar electrodes have been extensively studied using techniques such as XPS, SEM, and EIS [39,40] The active surface areas of GCE, Ch/GCE, and Ch-G/GCE were evaluated using CV with K_4_Fe(CN)_6_ in 0.05 mol/L KCl, and the MR activity was evaluated using CV and SWV. The voltammograms are shown in Figure 1. The results in Figure 1A show that the Ch-G/GCE-modified electrode increased the anodic and cathodic currents for the K_4_Fe(CN)_6_ solution (Figure 1A, curve c). There is also an increase in the background current compared to the unmodified electrode. These results show that the presence of chitosan and graphene on the GCE surface increases the surface area. On the other hand, the ∆V values for the electrodes GCE (curve a), Ch/GCE (curve b), and Ch-G/GCE (curve c) were 0.19, 0.17, and 0.20 V, respectively. These results show that the reversibility of the K_4_Fe(CN)_6_ solution remained almost the same after the modification. Therefore, the energy needed to oxidize MR is almost the same in the three electrodes. Chitosan is not a conductive polymer, and the increase in anodic peak currents in Figure 1A, curve b, can be caused by hydrogen bridges, hydrophobic interactions, and electrostatic interactions [18].

To calculate the active area value in cm^2^, and to determine the area increase with the modified and unmodified electrodes, anodic and cathodic peak currents for the same K_4_Fe(CN)_6_ solution were studied via CV, varying the scan rates between 0.02 and 0.12 V/s with Ch/GCE and Ch-G/GCE. These results were compared with those of unmodified GCE. The results for Ch-G/GCE are shown in Figure 1B,C. In the three electrodes, it was clearly observed that anodic peak currents increased proportionally with an increasing scan rate of between 0.02 and 0.10 V/s, but there was a slight increase with Ch-G/GCE. The linear relation for the anode and cathode currents (I_pa_ and I_pc_) versus the scan rate (ʋ)½ was plotted for the three electrodes, and the values of the slopes were substituted into the Randles–Sevcik equation [40,44]:I_p_ = 2.69 × 10^5^ n ^3/2^ AD^1/2^ ν ^½^
*C*
to obtain the value of the active surface area in cm^2^, where I_p_ is the anodic peak current, n is the number of electrons (*n* = 1), D is the diffusion coefficient for K_4_Fe(CN)_6_ (7.6 × 10^−6^ cm^2^s^−1^), ν^1/2^ is the scan rate, and *C* is the ferricyanide concentration in mol/cm^3^.

Table 1 summarizes the area values obtained for the three electrodes, where an increase was observed for Ch/GCE and Ch-G/GCE compared to the unmodified GCE. This may be the cause of the increase in current with the electrode modified with chitosan and graphene. In previous reports, where the same electrodes were developed, but with different methodologies, and where the surface of the electrode modified only with graphene was evaluated, this value was similar to the value calculated for the electrode modified only with chitosan of 0.05 cm^2^. On the other hand, for the electrode modified with graphene and chitosan, the value obtained was slightly lower than that in a previous report, which was 0.16 cm^2^ [40]. Moreover, these results are very similar to previous reports on the use of electrodes with metal oxides and metal nanoparticles [44,45,46].

The electro-oxidation of MR usually involves the transfer of one and two electrons with similar protons, generating one of the aromatic rings between 0.2 and 0.5 V [9,12]. In this study, MR activity using the three electrodes was studied. The results are shown in Figure 2. Anodic peak currents for MR were observed between 0.40 and 0.60 V for CV and SWV, but there was a large increase in current of almost 80%, with Ch-G/GCE at 0.5 V per CV and 0.48 V via SWV. These results are shown in Figure 2A,B, curves c. This significant increase in the MR anode current peak is possibly due to a synergistic effect of graphene that allows for a higher rate of charge transfer [13] and an increase in the MR concentration on the electrode surface, due to increases of the active area surface, which is a very similar effect to that observed in previous reports, where it was used to detect rutin [40,41]. Moreover, when electrodes modified with nanoparticles of metals such as Ni, Zn, and Mo were used, the morin oxidation peak was observed with potential values of less than 0.5 V [10,12,13]. With electrodes modified with graphene and silver oxide, the morin oxidation peak was observed at a value greater than 0.5 V [16]. This indicates that the modified electrode in this report, although it is simpler to produce compared to electrodes that require less energy for MR oxidation, does not require as much energy, compared to other more complex electrodes. On the other hand, SWV allows us to observe the activity of MR at concentrations almost 100 times lower than that of CV. The results of the increase in the anode peak currents for MR are summarized in Table 2 for CV and SWV, where it is observed that the electrode modified only with chitosan does not significantly affect the activity of morin. Therefore, the function of chitosan only allows graphene to be immobilized on the glassy carbon surface.

### 3.2. Evaluation of the Mass Transfer Process for MR with Ch-G/GCE

CV was used to study the mass transfer processes for 0.12 mmol/L MR by varying the scan rate between 0.02 and 0.10 V/s with a Ch-G/GCE. The results are shown in Figure 3. A linear increase in anodic peak currents was observed as the scan rate increased from 0.02 to 0.10 V (Figure 3A), but the plot of the log ʋ (V/s) vs. log I_p_ (µA) has a slope value of far from one. This indicates there is no absorption-controlled process. However, for MR, the I_pa_ vs. scan rate (ʋ) ^½^ was linear, with a correlation coefficient R^2^ of 0.989 (Figure 3B), indicating that the process is diffusion controlled. To prove this result, with the same values, the plot of log I_pa_ vs. log ʋ was also prepared (Figure 3C), where the slope of the linear equations was 0.42. This indicates the processes are diffusion controlled for MR, because these slope values were close to the theoretical value of 0.5. Normally, using electrodes modified with other materials, such as carbon nanotubes, metal oxides, and cation nanoparticles [8,9,10,11,13], the process for the oxidation of MR is controlled by adsorption. Then, the process controlled by diffusion favors an increase in the useful life of the electrode, since it allows for the total cleaning of the electrode, reducing the memory effects of the previous measurements. On the other hand, the voltammograms in Figure 3A show that the potential values shift to more positive values with an increase in the scan rate, which indicates a process that is limited by reaction kinetics [11]. In this case, the plot of E (V) vs. log scan rate (mV/s) resulted in a linear regression equation, E(V) = 0.339 + 0.095 log ʋ (mV/s), with a correlation coefficient of 0.99. This slope value of close to zero indicates a very slow kinetic process that does not affect the rate of charge transfer for MR oxidation. On the other hand, the plot of curve I_pa_ (A) vs. (ʋ) ½ presents a slope value of 6.0 × 10^−5^ with 0.12 mmol/L MR. For Ch-G/GCE, an active area of 0.16 cm^2^ was obtained from the Randles–Sevcik equation, with a diffusion coefficient D for MR of 5.7 × 10^−6^ cm^2^s^−1^ and z = 2e^−^ [47]. This value is close to the reported in Table 1 for Ch-G/GCE. The shift of the maximum potentials of MR with the increase in the scan rate indicates that the irreversibility of the reaction was increased.

### 3.3. pH Effect on MR Using the Ch-G/GCE

The effect of the pH of the solution is important, because this substance has antioxidant properties due to the loss of H^+^ protons that can neutralize free radicals [48,49]. Therefore, the anodic peak current and anodic peak potentials based on the pH were analyzed. The pH was studied at 2.5, 3.0, 4.0, 5.5, 6.0, and 7.0 via SWV using Ch-G/GCE and 6.0 µmol/L MR. The results in Figure 4A show that the anodic peak potential shifted to more positive potential values at more acidic pH values. The potential E(V) versus the pH was plotted (Figure 4B), and the linear equation was E_p_(V) = 0.571–0.043 pH. This result indicates that protons (H^+^) are involved in the oxidation of MR, and a slope value near 0.059 indicates that H^+^ = e^−^ are the same; normally, a ratio of 2:2 has been reported in the MR reaction. These results are similar to those previously reported for MR using electrodes modified with different materials [9,10,13]. On the other hand, the highest oxidation current was observed at pH 3.0. It is possibly caused by the pKa values for morin, which are between 4.7 and 10.7 [8]. Thus, at pH 3.0 the current is higher, because at that pKa value, all of the protons are present. This pH value was chosen for further studies.

### 3.4. Calibration Curve, Sensitivity, and Detection Limit

Net SWV currents were used under optimized conditions: a pH of 3.0 (phosphate-buffered solution), 0.0 V by 30 s in the accumulation step, a frequency of 15 Hz, an amplitude of 0.05 V, and a potential increment of 0.01 V in the scan step. These parameters were calculated with the univariate method, and those with the highest currents and symmetrical peak shapes were taken as being optimal. These currents were used to measure the limit of detection and sensitivity of the new method with the Ch-G/GCE for MR, with 20.0 µL being added consecutively from a standard solution of 0.5 mmol/L with 9.0 mL of ultrapure water and 0.250 mL of PBS in the electrochemical cell. SW voltammograms and calibration curves are shown in Figure 5. The detection limit was 3(S_x/y_)/slope, the limit of quantification was 10(S_x/y_)/slope, and the sensitivities were 0.30 μmol/L, 1.0 μmol/L, and 0.33 μA/μmol/L for MR. This value was calculated with the first 10 points in Figure 5B. Table 3 shows a summary of some previous MR studies using electrodes modified with metal ion coordination complexes, metal ion nanoparticles, carbon nanotubes, and polymers. In the results of Table 3, it is clearly observed that the most sensitive sensors used electrodes modified with nanoparticles of metal ions and coordination complexes, followed by some that used carbon nanotubes, and to a lesser extent, but still sensitive below 1.0 µmol/L, with polymers. On the other hand, these electrodes are not easy to manufacture. In this work, an easier-to-manufacture sensor is reported, which is equally sensitive compared to others that use graphene and carbon nanotubes combined with the coordination complexes. In addition, the linear concentration range was close to other previously reported values (see Table 3), where carbon nanotubes were mainly used.

### 3.5. Interference Study and Reproducibility

Some substances that can interfere with MR detection, such as quercetin (QC), rutin (RT), and catechin (CT), were evaluated. CT did not show activity with Ch-G/GCE. QC showed a much lower activity with the electrode at less positive potential values between 0.1 and 0.2 V, a result that was very similar to that reported using AgNPs-G/GCE [16]. On the other hand, RT showed activity with the electrode at a potential very close to that of MR. The result of this activity, which can cause considerable interference with the MR quantification, is shown in Figure 6. The separation of both anodic peak currents is 60 mV. Furthermore, this interference was evaluated with a real sample. The results are shown in Figure 6B. It is observed that the sample has an anodic peak current signal at 0.46 V (Figure 6B curve a), and when a known amount of MR is added, the anodic peak current increases to the same potential value (Figure 6B curve b). Then, the same amount of RT is added (Figure 6B curve c). In the results, no increase is observed at 0.46 V, but a broadening of the signal is observed at 0.54 V. These results show that RT can cause a small interference with the MR signal at higher concentrations compared to those detected for MR in this type of sample, such as coffee and tea. This is because the activity for this sensor is higher for MR compared to RT, as shown in Figure 6A, where the concentration of RT is double that of MR. Previous reports of RT using similar electrodes, but with different methodologies, observe the RT anodic peak current signal at 0.5 V [40,41]. With this new methodology, the oxidation signal changes to more positive potential values. In addition, the RT signal decreases considerably with an increasing pH of between 5.0 and 6.0, and although the MR signal shifts to 0.25 V, it only decreases by 20%, which allows it to be sensitive.

The relative standard deviation (%RSD) was calculated from the measurements obtained for 5.0 µmol/L MR using three different electrodes via SWV. The standard deviation of the anodic peak current was 1.8%. The results show that the sensor was reproducible using different modified electrodes prepared in the same way. Moreover, to validate the accuracy of the new method, two water samples doped with two known amounts of morin (2.5 and 5.0 µmol/L) were analyzed. The samples were analyzed using Ch-G/GCE, and the results obtained were 2.2 and 5.4, obtaining % recovery values of 88.0 and 108%, respectively.

### 3.6. Analytical Applications

The utility of the new electroanalytical method with Ch-G/GCE was applied in the detection of MR in samples of coffee and aromatic chamomile beverages. The results are summarized in Table 4, and the voltammograms and calibration curves of the aromatic chamomile beverage sample are shown in Figure 7. The results showed that the sample matrix did not affect the potential values for MR, and there was only a small change of 20.0 mV. On the other hand, the value of the slope of this curve was 0.407 C_MR_. This value is close to the value of the slope inserted in Figure 5B. This value also confirms that the effect of the sample matrix was not considerable. These detected values were higher than those detected in wine grapes [16], chocolate, tea [50], and plant and fruit tissue extract [8,12], but similar to those detected in fruits such as mulberry [9]. In addition, the values detected are within the normal range detected for samples of plant origin [51]. The disturbance observed in the blank is the product of small differences in the electrical supply.

## 4. Conclusions

In summary, the new application for glassy carbon electrodes modified with chitosan and graphene to detect MR was proven to be sensitive, selective, and reproducible. The results showed that the increase in the surface area improved the sensitivity, and that the synergy between chitosan and graphene favored selectivity in the presence of other flavonoids. The detection limit and sensitivity were 0.30 μmol/L and 0.33 μA/μmol/L, respectively. It was observed that chitosan does not significantly affect the MR activity, because the anode peak current does not increase in the presence of chitosan. Additionally, the presence of graphene considerably improves the MR activity, due to the large increase in the anode peak current. Moreover, the real samples did not need treatment to minimize the effect of the matrix. The values detected for MR in the real samples of this report are within the normal range detected with other electrodes and other types of samples. This new report also contributes to the development of easy-to-develop and low-cost sensors of environmental, biological, and food interest.

## Figures and Tables

**Figure 1 sensors-22-07780-f001:**
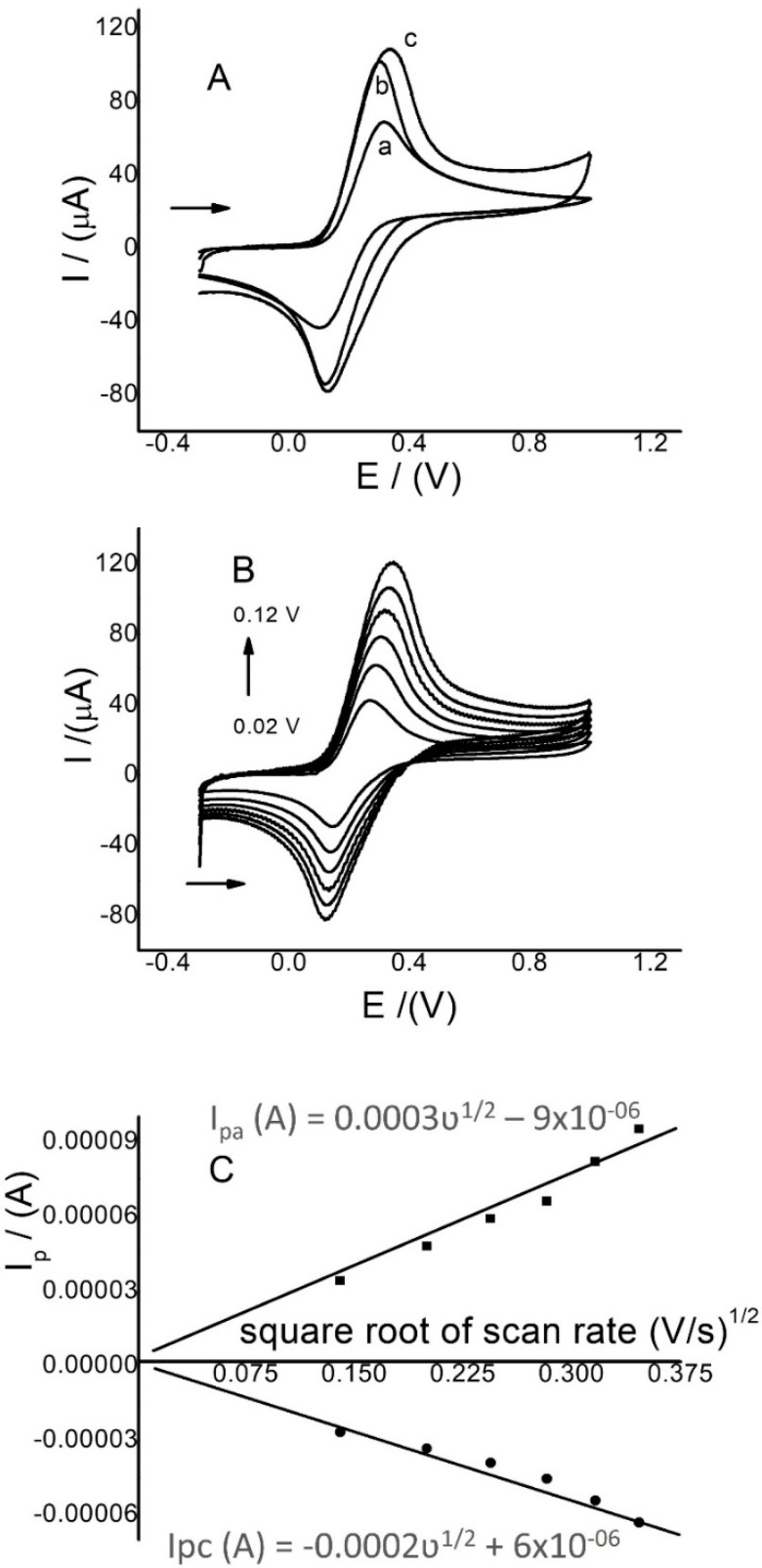
(**A**) CVs of K_4_Fe(CN)_6_ (5.0 mmol/L) with GCE (curve a), Ch/GCE (curve b), and Ch-G/GCE (curve c) at 0.10 V/s; (**B**) CVs of K_4_Fe(CN)_6_ (5.0 mmol/L) at scan rates between 0.02–0.12 V/s with Ch-G/GCE; and (**C**) plot of I_p_ (**A**) versus ʋ^1/2^ (V/s).

**Figure 2 sensors-22-07780-f002:**
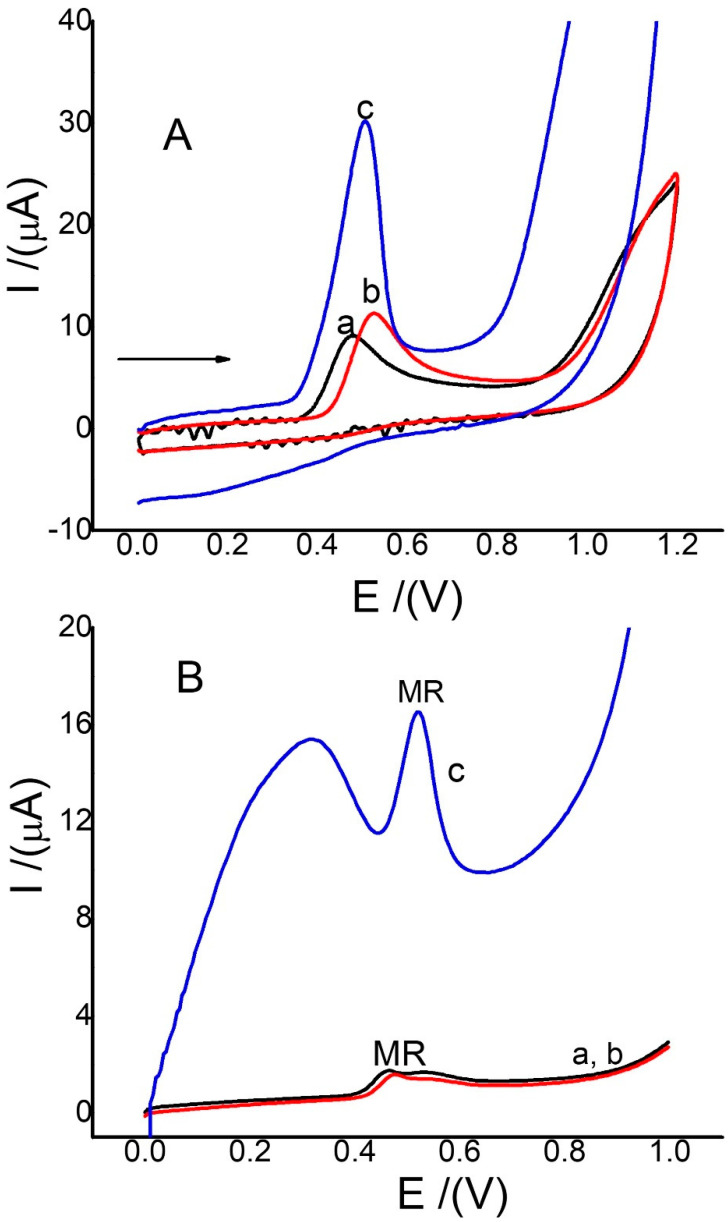
(**A**) CVs of MR (0.12 mmol/L) using GCE (curve a, black), Ch/GCE (curve b, red), and Ch-G/GCE (curve c, blue) at a scan rate of 0.1 V/s; and (**B**) SWVs of MR (6.0 µmol/L) with GCE (curve a, black), Ch/GCE (curve b, red,), and Ch-G/GCFE (curve c, blue). Conditions: pH 3.0 (phosphate-buffered solution) at 0.0 V for 30 s, frequency 15 Hz, amplitude 0.05 V, and potential increment 0.01 V.

**Figure 3 sensors-22-07780-f003:**
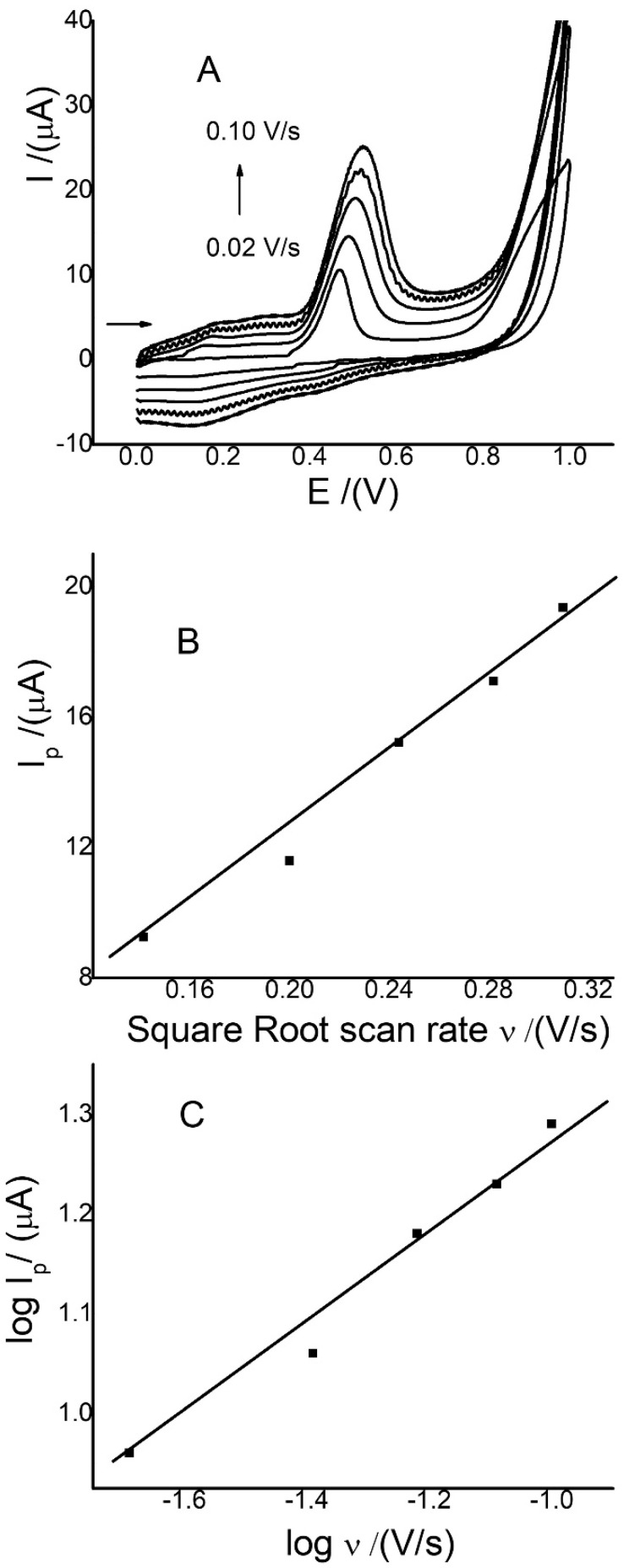
(**A**) CVs for 0.12 mmol/L MR between 0.02 and 0.12 V/s; (**B**) plot of (ʋ)^1/2^ versus I_p_ (µA) for 0.12 mmol/L MR; and (**C**) plot of log I_p_ (µA) versus log of ʋ (V/s), using Ch-G/GCE at pH 3.0.

**Figure 4 sensors-22-07780-f004:**
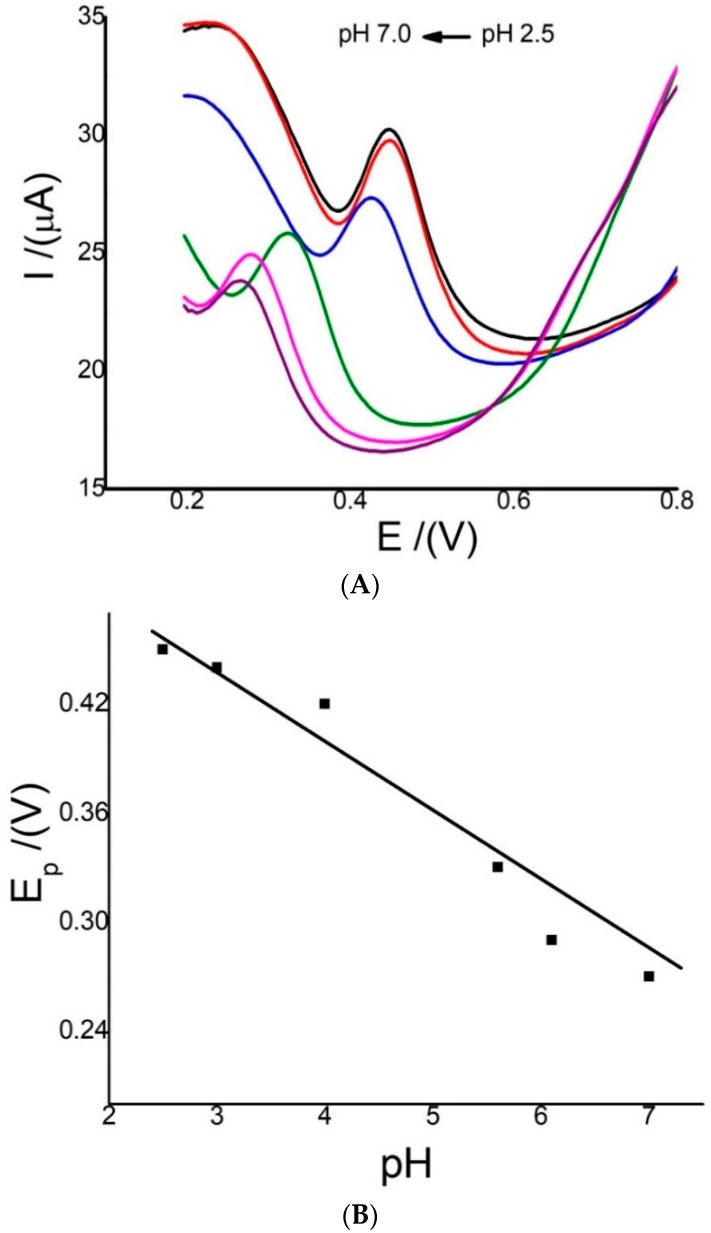
(**A**) SWVs of 0.12 mmol/L MR at pH values of 2.5 (black line), 3.0 (red line), 4.0 (blue line), 5.5 (green line), 6.0 (fuchsia line), and 7.0 (violet line); and (**B**) the effect of pH on E_p_(V) using Ch-G/GCE. Conditions: similar to those in Figure 2B.

**Figure 5 sensors-22-07780-f005:**
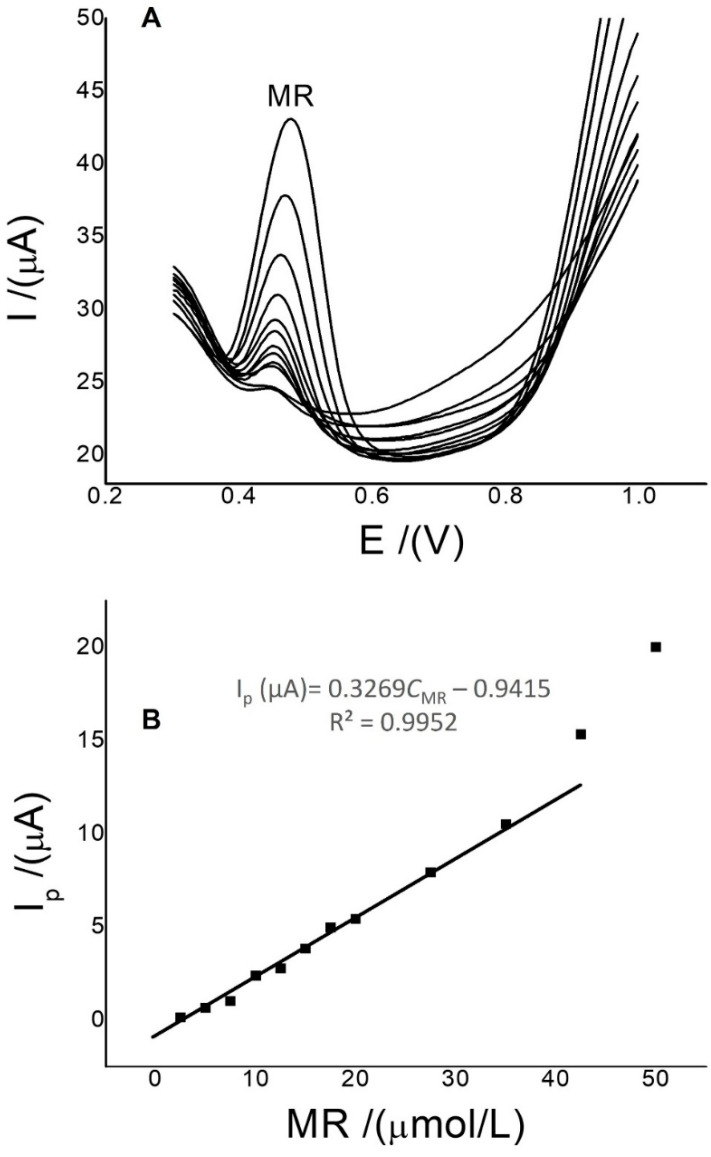
SWVs (**A**) and a calibration curve of MR (**B**) using Ch-G/GCE. Conditions: pH 3.0 (phosphate-buffered solution) at 0.0 V for 5 s. Frequency, 15 Hz; amplitude, 0.05 V; and potential increment, 0.01 V.

**Figure 6 sensors-22-07780-f006:**
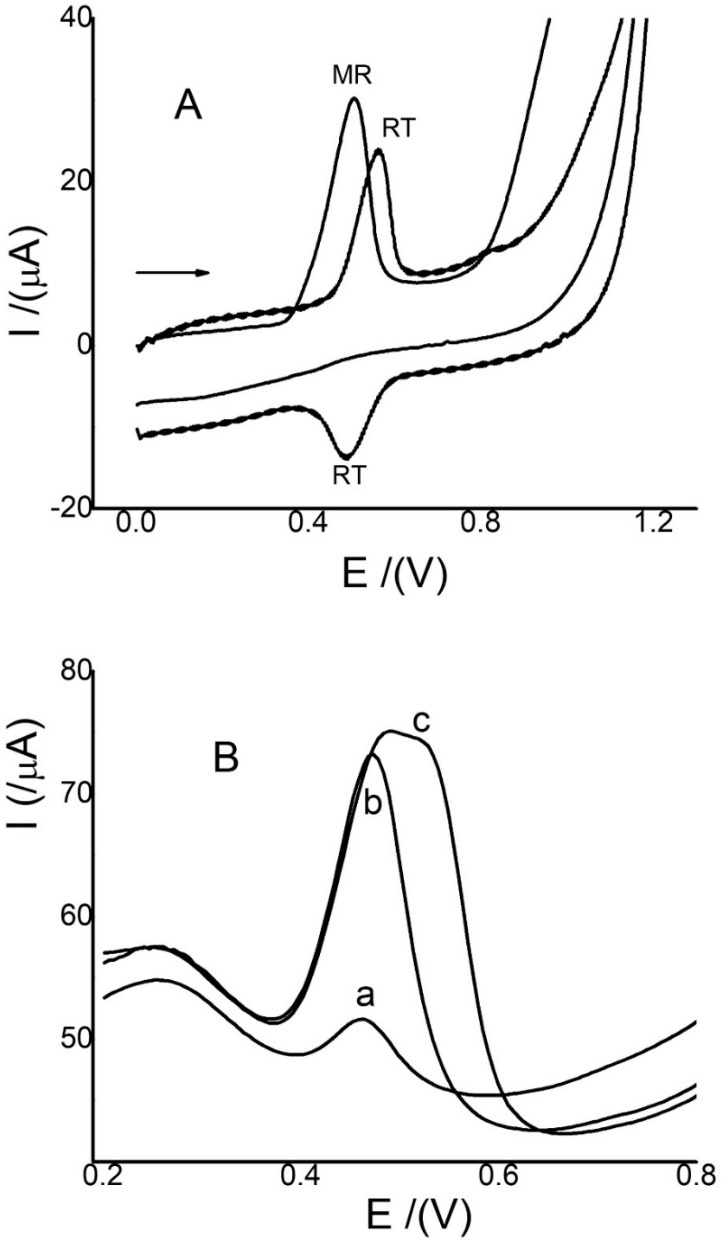
(**A**) CVs of MR (0.12 mmol/L) and RT (0.24 mmol/L), with Ch-G/GCE using 0.1 V/s and phosphate-buffered solution with a pH of 3.0; and (**B**) SWVs of coffee sample (curve a), MR 20.0 µmol/L (curve b), and RT 40.0 µmol/L with Ch-G/GCE. (curve c) Conditions: pH 3.0 (PBS) at 0.0 V for 30 s, 15 Hz, and pulse amplitude 0.05 V.

**Figure 7 sensors-22-07780-f007:**
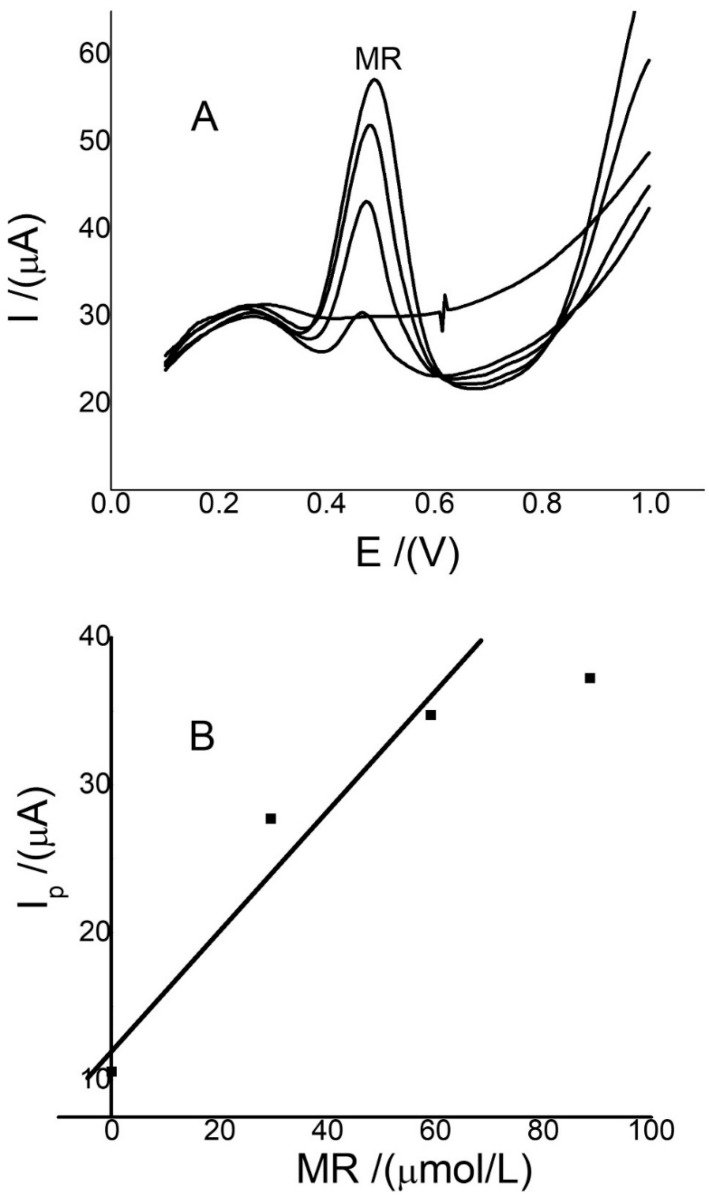
SWVs (**A**) and calibration curve for MR; (**B**) for the samples of aromatic chamomile beverages using Ch-G/GCE. Concentration was calculated with the first three points. The same conditions were used as in. Figure 6A.

**Table 1 sensors-22-07780-t001:** Surface active areas of *GCE*, *Ch/GCE*, and *Ch-G/GCE* electrodes.

Electrode	Slope (I_p_/*ν* ^½^*)*		Area (cm^2^)		Total Area (cm^2^)
	I_pa_	I_pc_	I_pa_	I_pc_	
*GCE*	0.0001	1.5 × 10^−5^	0.010	9.6 × 10^−4^	0.011
*Ch/GCE*	0.0002	0.0001	0.032	0.021	0.053
*Ch-G/GCE*	0.0003	0.0002	0.081	0.051	0.140

**Table 2 sensors-22-07780-t002:** Anodic peak currents for MR with *GCE*, *Ch/GCE*, and *Ch-G/GCE* by CV and SWV.

Electrode	CV		SWV		%I_pa_ Increased	%I_pa_ Increased
	E(V)	I_pa_(µA)	E(V)	I_pa_(µA)	CV	SWV
*GCE*	0.47	6.5 ± 0.2	0.47	1.00 ± 0.01		
*Ch/GCE*	0.46	9.2 ± 0.4	0.47	0.97 ± 0.03		
*Ch-G/GCE*	0.50	25 ± 0.1	0.52	4.80 ± 0.01	75.0	79

**Table 3 sensors-22-07780-t003:** Detection of MR with modified electrodes between 2014 and 2021.

Electrode Composite	Application	Range Linear (µmol/L)	DoL (µmol/L)	Ref.
CNT/IL	Vegetables	0.1–30.0	0.05	[8]
CNT/SF	Mulberry	0.01–0.09	0.02	[9]
MoS_2_/G			0.39	[10]
CNF/Tb_2_Se_2_	Guava leaves	0.1–10.0	0.60	[11]
NH_2_-CNT/ZnO	Fruits	0.2–803	0.02	[12]
NiPc/CPE	Human urine, foods	0.006–250	0.002	[13]
Co(NH_3_)_6_-G/GCE	Fruits	0.008–72.0	0.008	[14]
PEDT–Au/G/GCE	Human serum	1.0–150.0	0.83	[15]
AgNPs-G/GCE	Grape wine	0.003–1.0	0.003	[16]
Ch-G/GCE	Caffe	0.9–50.0	0.30	This work

**Table 4 sensors-22-07780-t004:** Analysis of real samples (*n* = 3).

Samples	MR (µmol/L)	
	Found	%RSD
Coffee	18.5-	0.03
* Aromatic chamomile beverage	30.5	0.01
Natural fruit juice	11.6	0.01

* Voltammograms are shown in Figure 7.

## Data Availability

All data will be provided upon reasonable request to the corresponding author.

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
