# Peer review of "Electrochemical Determination of Morin in Natural Food Using a Chitosan–Graphene Glassy Carbon Modified Electrode"

_sensors, 2022, doi:10.3390/s22207780_

Round 1
Reviewer 1 Report (Previous Reviewer 1)
The article can be acceptable in the present form
Author Response
Thanks for all the suggestions that help improve the quality of the manuscript

Reviewer 2 Report (Previous Reviewer 3)
The conclusions drawn from the experimental results are either not correct or contradict the methodology used in electroanalytical studies:
1. The upper limit of the concentration range of linear electroanalytical response of the investigated system is clearly about 35 micromols and not 50 micromols (Fig. 5) as claimed (Table 3) .
2. At the claimed lower limit (0.9 micromols) of the concentration range of linear electroanalytical response, the value of the current (acc. to the calibration equation in Fig. 5) will be -0.64729, i.e negative ! In fact for all concentrations below 2.88 micromols the current values will be negative which is of course not acceptable. Therefore the claimed lower limit of the concentration range of linear electroanalytical response is also not correct.
3. The real sample analysis study is made by three standard additions of 30 micromols each. Once the concentration range of linear electroanalytical response of the system is limited to only 35 micromols, this contradicts the common sense. This is not how the standard addition method works at all.
4. The comparison made with other electrode materials investigated for morin electrochemical determination show that there is no progress made with the present study.
Finally, there is obviously a problem with the behavior of the electrode in the supporting electrolyte (base line shift as the authors explained). In such a case they should better work to solve this problem before trying to publish unacceptable data.
The technical quality of the figures was not improved - the position of the numbers on the X and Y -axis is not clearly indicated (missing side bars) which impedes the reading of the values on the plots. There are standards for publishing of scientific results which are not kept in this manuscript.
Author Response
The conclusions drawn from the experimental results are either not correct or contradict the methodology used in electroanalytical studies:
Reply: We extract concussions based on the results, not so much on the methodology. the only conclusion based on the methodology was for the treatment of the sample, which did not require further treatment
- The upper limit of the concentration range of linear electroanalytical response of the investigated system is clearly about 35 micromols and not 50 micromols (Fig. 5) as claimed (Table 3) .
Reply: we indicate the 50.0 uM data because it was in the measurements, although the linearity had been lost a little. the change was made in table 3
- At the claimed lower limit (0.9 micromols) of the concentration range of linear electroanalytical response, the value of the current (acc. to the calibration equation in Fig. 5) will be -0.64729, i.e negative ! In fact for all concentrations below 2.88 micromols the current values will be negative which is of course not acceptable. Therefore the claimed lower limit of the concentration range of linear electroanalytical response is also not correct.
Reply: the observation is correct. That is why the value of the intercept in Ip is negative, but it should be noted that it is not so far from zero. the only electrodes that the intercept approaches zero are HMDE. solid electrodes move away a little, even in some cases the value is positive, but close to zero. that does not indicate that the measurements are wrong, it is a deviation from ideality. therefore the detection limit was calculated with the standard error of the curve
- The real sample analysis study is made by three standard additions of 30 micromols each. Once the concentration range of linear electroanalytical response of the system is limited to only 35 micromols, this contradicts the common sense. This is not how the standard addition method works at all.
Reply: the real sample is diluted, a single dilution with the total volume of the cell, that is why the current is lower, for the final calculation this dilution factor is considered, as it is developed in any process
- The comparison made with other electrode materials investigated for morin electrochemical determination show that there is no progress made with the present
Reply: this comment is true, but we never try to improve or be the best sensor, what we indicate in the objective is the development of a sensor that is easier to manufacture compared to other electrodes, but equally sensitive and versatile
Finally, there is obviously a problem with the behavior of the electrode in the supporting electrolyte (base line shift as the authors explained). In such a case they should better work to solve this problem before trying to publish unacceptable data.
The technical quality of the figures was not improved - the position of the numbers on the X and Y - axis is not clearly indicated (missing side bars) which impedes the reading of the values on the plots. There are standards for publishing of scientific results which are not kept in this manuscript.
Reply: the background current problem when the real samples were used are normal, the important thing is that the signal is not affected, in this case the linearity, that happens because the optimization and characterization work is done with ultra pure water and the samples real changes that medium and we consider that it is not necessary to try to evaluate this interference if the change in potential is small and the current remains proportional to the concentration.
on the other hand, the authors indicated that the statistical program is with this application and we cannot change it unless we get updates and we do not have the resources at this time. therefore, the authors have tried to include all the current and potential values in tables so that it is not necessary to locate them in the figures.
Reviewer 3 Report (New Reviewer)
This report presents a new application for the chitosan-graphene glassy carbon electrode system in the determination of the hydroxyflavonoid morin by square-wave voltammetry. The work is quite interesting and the analytical method was well conducted. However, there are some doubts and corrections to be made. Therefore, I recommend that a major review be carried out so that the manuscript can be considered for publication in Sensors.
General comments:
1. According to the IUPAC definition, "graphene is a single carbon layer of graphite structure, describing its nature by analogy of a polycyclic aromatic hydrocarbon of quasi-infinite size". The authors used “oxide graphene” and “reduced graphene oxide”. See more about the nomenclature of these materials in Carbon, 65 (2013) 1-6.
2. Modified electrodes need to be characterized morphologically. I recommend adding scanning electron microscopy images for GCE, CS/GCE and CS-G/GCE.
3. Since chitosan is soluble in an acidic medium, how do the authors guarantee that at pH 3.0 (optimized) there was no leaching of the film in the GCE?
4. The Abstract needs to be improved by describing more results of the work.
5. Use the correct representations: K4[Fe(CN)6], K3[Fe(CN)6], [Fe(CN)6]4−, [Fe(CN)6]3−.
6. Square wave voltammetry has three experimental parameters: amplitude, frequency and potential increment. Supply correctly.
Specific comments:
1. Introduction:
a. Lines 41-42. Potential is a relative value. Thus, the reference electrode (e.g. vs. Ag/AgCl, vs. SHE, vs. SCE) needs to be indicated.
b. Line 63. Chitosan has the ability to form stable films on GCE. This is an important skill to prevent leaching of the modified agent into the supporting electrolyte. Add the sentence “Chitosan has the ability to form stable films on surfaces". Add the Microchemical Journal, 167 (2021) 106297 and Talanta, 252 (2023) 123836 references to validate this information.
2. Materials and methods:
a. Do not use the acronym PBS for "phosphate-buffered solution". PBS is the accepted acronym for "phosphate-buffered saline" which contains 0.9% NaCl to warrant physiological ionic strength. See the Sigma Aldrich catalog (product P5368), for example. Also, see: https://en.wikipedia.org/wiki/Saline (medicine). Unfortunately, PBS is often wrongly used as an acronym for "phosphate buffer(ed) solution" in the literature but this is wrong and can cause confusion. Does the buffer employed by the authors really contain 0.9% NaCl (or other electrolytes such as MgCl2)? If yes, please specify. See: https://en.wikipedia.org/wiki/Phosphate-buffered_saline.
b. Informs the city and country where the samples were acquired.
c. Standardize the description of the reagents and equipment: model (company, country).
d. Line 107. How do the authors guarantee that the application of a constant potential of −1.0 V did not generate H2 evolution?
3. Results and discussion:
a. Figure 1A. Provide the peak-to-peak separation values of the [Fe(CN)6]3−/4− probe at the three different electrodes. Discuss the reversibility of the reaction.
b. Currently the acronym for the number of electrons is z. Update the terminologies for the electrochemical methods according to new IUPAC recommendations. See the information in Pure and Applied Chemistry, 92 (2020) 641–694.
c. Lines 172-173. The mechanism of morin oxidation needs to be clarified. Is it one or two electrons?
d. Lines 214-216. The sentence needs to be validated with a reference.
e. Lines 217-219. The shift of the peak potentials of a species with increasing scan rate indicates the irreversibility of the reaction.
f. Section 3.3. Add a discussion about the pKa of morin.
g. How was the detection limit calculated? Also, inform the limit of quantification.
h. Line 308. Provides the recovery values (not the RE%). The recovery assay responds to the percentage of recovery of the analyte added to a sample. The recovery percentage should ideally be 100%. See the information in Trends in Analytical Chemistry, 26 (2007) 227-238.
i. Figure 7B. Plot the calibration curve together with the standard addition curve to compare the parallelism of both. If they are parallel, there was no interference from the sample matrix components.
j. The correct is “anodic peak current”, not anode.
k. Line 238. “The potential E(V) versus the pH (Ê‹)½ was plotted”. (Ê‹)½ ??
Author Response
This report presents a new application for the chitosan-graphene glassy carbon electrode system in the determination of the hydroxyflavonoid morin by square-wave voltammetry. The work is quite interesting and the analytical method was well conducted. However, there are some doubts and corrections to be made. Therefore, I recommend that a major review be carried out so that the manuscript can be considered for publication in Sensors.
General comments:
- According to the IUPAC definition, "graphene is a single carbon layer of graphite structure, describing its nature by analogy of a polycyclic aromatic hydrocarbon of quasi- infinite size". The authors used “oxide graphene” and “reduced graphene oxide”. See more about the nomenclature of these materials in Carbon, 65 (2013) 1-6.
Reply: commercial graphene from sigma-aldrich was used, which comes as graphene oxide because it contains some functional groups, the process developed in the methodology applying a negative potential is done to try to eliminate these groups, that is why it is called in many cases reduced graphene, the nomenclature given in this work as G is only to abbreviate the name of the modified electrodes a little and to make its representation easier. It is not about creating a new nomenclature, normally many authors indicate it as G.
- Modified electrodes need to be characterized I recommend adding scanning electron microscopy images for GCE, CS/GCE and CS-G/GCE.
Reply: this system has already been characterized by these techniques and the authors considered that there would be nothing new in this characterization. Also, we would like to complete this study, but we are a public university that does not have all the instruments, that is why the work is mainly about the development of an electroanalytical sensor and not the study of the physicochemical activity of a surface
- Since chitosan is soluble in an acidic medium, how do the authors guarantee that at pH 0 (optimized) there was no leaching of the film in the GCE?
Reply: the authors have reported several works with chitosan and this has the advantage that after it forms the film it does not dissolve easily in an acid medium, on the other hand, it is soluble in acetic acid and in the pH was used another acid.
on the other hand, the reproducibility of section 3.5 evidence that decomposition of the chitosan film does not occur. This was added in the text
- The Abstract needs to be improved by describing more results of the
Reply: The abstract was improved
- Use the correct representations: K4[Fe(CN)6], K3[Fe(CN)6], [Fe(CN)6]4−, [Fe(CN)6]3−.
Reply: the correct representations was added
- Square wave voltammetry has three experimental parameters: amplitude, frequency and potential Supply correctly.
Reply: parameters were added correctly.
Specific comments:
- Introduction:
- Lines 41-42. Potential is a relative Thus, the reference electrode (e.g. vs. Ag/AgCl, vs. SHE, vs. SCE) needs to be indicated.
Reply: the reference electrode was added
- Line 63. Chitosan has the ability to form stable films on GCE. This is an important skill to prevent leaching of the modified agent into the supporting Add the sentence “Chitosan has the ability to form stable films on surfaces". Add the Microchemical Journal, 167 (2021) 106297 and Talanta, 252 (2023) 123836 references to validate this information.
- Materials and methods:
- Do not use the acronym PBS for "phosphate-buffered solution". PBS is the accepted acronym for "phosphate-buffered saline" which contains 9% NaCl to warrant physiological ionic strength. See the Sigma Aldrich catalog (product P5368), for example. Also, see: https://en.wikipedia.org/wiki/Saline (medicine). Unfortunately, PBS is often wrongly used as an acronym for "phosphate buffer(ed) solution" in the literature but this is wrong and can cause confusion. Does the buffer employed by the authors really contain 0.9% NaCl (or other electrolytes such as MgCl2)? If yes, please specify. See: https://en.wikipedia.org/wiki/Phosphate-buffered_saline.
Reply: It is great information. We do not know this information since normally many authors indicate this symbol. The solution used does not contain NaCl. changes were made to the text
- Informs the city and country where the samples were
Reply: the data was added to the text
- Standardize the description of the reagents and equipment: model (company, country).
Reply: the data was added to the text
- Line How do the authors guarantee that the application of a constant potential of −1.0 V did not generate H2 evolution?
Reply: the authors do not guarantee this process, bubble formation was observed on the surface of the electrode, possibly due to this process. The time of 60 s allows this process not to affect the activity of the sensor
- Results and discussion:
- Figure Provide the peak-to-peak separation values of the [Fe(CN)6]3−/4− probe at the three different electrodes. Discuss the reversibility of the reaction.
Reply: The data was added to the text
- Currently the acronym for the number of electrons is z. Update the terminologies for the electrochemical methods according to new IUPAC See the information in Pure and Applied Chemistry, 92 (2020) 641–694.
Reply: in many formulas e- is symbolized as n for that reason it was put like this. In other parts of the manuscript it was changed to z
- Lines 172-173. The mechanism of morin oxidation needs to be Is it one or two electrons?
Reply: the results showed that the amount of H+ and e- are equal. Normally the ratio of 2:2 is the most published.Changes were made to the manuscript
- Lines 214-216. The sentence needs to be validated with a
Reply: previous changes made to the manuscript caused the lines to move and the authors do not know to which part of the manuscript the reviewer suggests adding a reference
- Lines 217-219. The shift of the peak potentials of a species with increasing scan rate indicates the irreversibility of the reaction.
Reply: is a great observation that was added in the text
- Section 3. Add a discussion about the pKa of morin.
Reply: Reply: The data was added to the text
- How was the detection limit calculated? Also, inform the limit of
Reply: was calculated as 3(sx/y)/slope and limit of quantification was added as 10(sx/y)/slope
- Line 308. Provides the recovery values (not the RE%). The recovery assay responds to the percentage of recovery of the analyte added to a sample. The recovery percentage should ideally be 100%. See the information in Trends in Analytical Chemistry, 26 (2007) 227-238.
Reply: The data was added to the text
- Figure Plot the calibration curve together with the standard addition curve to compare the parallelism of both. If they are parallel, there was no interference from the sample matrix components.
Reply: the authors added the slope value of this calibration curve and compared it with the slope value of the calibration curve from section 3.4 to check if there was any effect of the sample matrix
- The correct is “anodic peak current”, not
Reply: anodic peak currents was completed
- Line 238. “The potential E(V) versus the pH (Ê‹)½ was plotted”. (Ê‹)½ ??
Reply: (Ê‹)½ was retired
Reviewer 4 Report (New Reviewer)
In this paper, Chitosan-graphene glassy carbon modified electrode was employed for the electrochemical determination of morin in natural food. The paper is well presented or written. Authors have focused completely electrochemical techniques for characterization of the electrode and detection of morin. However, the authors required to revise the manuscript before publication in this journal.
Comments:
1. Last sentence of the introduction is not enough to explain what author has done in this manuscript. Briefly need to explain what research work is conducted and how they performed the experiments.
2. Chitosan is non-conducting polymer, but the peak current (Fig. 1A) is higher than bare GCE? it should be explained.
3. In Figure 3, it should be square root of scan rate not just only scan rate. And several places there is a confusion between scan rate and square root of scan rate. Check it.
4. Conclusion should be rewrite by giving much more information such as detection limit, etc. Also, required to mention about future research work based on the proposed sensor.
Author Response
In this paper, Chitosan-graphene glassy carbon modified electrode was employed for the electrochemical determination of morin in natural food. The paper is well presented or written. Authors have focused completely electrochemical techniques for characterization of the electrode and detection of morin. However, the authors required to revise the manuscript before publication in this journal.
Comments:
- Last sentence of the introduction is not enough to explain what author has done in this manuscript. Briefly need to explain what research work is conducted and how they performed the experiments.
Reply: changes were added to the text
- Chitosan is non-conducting polymer, but the peak current (Fig. 1A) is higher than bare GCE? it should be explained.
Reply: This comment is true, but the same authors of this new paper have explained this effect in previous reports, which has also been explained by other authors. this was included in the text
- In Figure 3, it should be square root of scan rate not just only scan rate. And several places there is a confusion between scan rate and square root of scan rate. Check it.
Reply: the error was corrected, also in figure 1
- Conclusion should be rewrite by giving much more information such as detection limit, etc. Also, required to mention about future research work based on the proposed sensor.
Reply: the conclusion was completed in the text
Round 2
Reviewer 3 Report (New Reviewer)
I understand the limitations of public universities and the scrapping of scientific research in South America.
If the authors wish, we can contribute in the future by collaborating with microscopic analyzes in future works.
Finally, the authors answered the questions and the manuscript was improved. Therefore, I recommend that the manuscript be accepted for publication in Sensors.
Reviewer 4 Report (New Reviewer)
The authors have improved the quality of the manuscript in the revised manuscript. Now the paper can be published in sensors.
This manuscript is a resubmission of an earlier submission. The following is a list of the peer review reports and author responses from that submission.
Round 1
Reviewer 1 Report
Journal Name: Sensors
Title: Electrochemical determination of morin in natural food using a chitosan-graphene glassy carbon modified electrode
In the current research, the authors used chitosan-graphene composite for electrochemical sensing morin. The authors also used the developed sensors to detect morine in real samples like coffee. This is very important research in the area of sensors and its scope matches the MDPI sensors journal. But the article needs some major revisions to get published in MDPI sensors.
1. Introduction section looks very weak authors need to compare electrochemical sensing of morin with other commonly used analytical methods.
1. Literature review is poor, the authors need to discuss the importance of the graphene surface morphology in electrochemistry I recommend authors make a separate paragraph on it here I am mentioning some of the articles the authors need to discuss them. It will be very much helpful for readers
a) https://pubs.acs.org/doi/10.1021/cr500023c
b) https://pubs.rsc.org/en/content/articlelanding/2016/cs/c6cs00136j
c) https://www.sciencedirect.com/science/article/pii/S0009261421009787
d) https://pubs.rsc.org/en/content/articlelanding/2018/nj/c8nj03679a/unauth
e) https://www.sciencedirect.com/science/article/pii/S1388248113003433
f) https://pubs.acs.org/doi/abs/10.1021/acs.jpca.6b08810
g) https://www.sciencedirect.com/science/article/pii/S001346861731527X
h) https://link.springer.com/book/10.1007/978-1-4471-6428-9
i) https://www.frontiersin.org/articles/10.3389/fnins.2020.594235/full
2. In At Fig.7 there is some disturbance in the blank and the capacitance of the blank voltammogram is quiet the authors need to explain the reason for this behavior.
3. LOD of the current electrode has to compare with the other previously published results.
4. The quality of the figures needs to be improved
5. The resistance of the modified electrodes needs to be calculated using EIS
6. What is the reason for taking such a wide potential? It looks like 0.2 to 0.8 V is sufficient
7. Conclusion looks very short it has to be improved
8. At Fig 6b potential is shifted positively what is the reason behind it ?
9. What are all the advantages of chitosan in modifying graphene composite needs to be discussed in brief.
Reviewer 2 Report
This paper proposed a new application of the chitosan-graphene glassy carbon modified electrode based electrochemical detection.but There are some problems. My detailed comments are as follows:
First,It is noted that your manuscript needs careful editing by someone with expertise in technical English editing paying particular attention to English grammar, spelling, and sentence structure so that the goals and results of the study are clear to the reader. only then can a proper review be performed.
There is at least one language logic error error in the manuscript, such as, in page5,line 162,“for”would be “in”. Please check the manuscript carefully.
Second, the significance of the paper is not expound sufficiently. The author need to highlight this paper's innovative contributions.And most citations is not rigorous.
Another obvious problem with this paper is data tagging can be clearer.For example,in page 4,figure(B), The CV curves of the unmodified glassy carbon electrode can be marked with different colors, which is convenient for readers to obtain effective information quickly.
Reviewer 3 Report
The core of this investigation is an electroanalytical study on the prospect to use a chitosan-graphene-modified GCE for electrochemical sensing of morin.
Exactly this part of the manuscripts has many deficiencies and fails to convince with the presented results.
In particular:
1. It is not clear how was obtained the calibration line (fig. 5 B) based on the data shown in fog. 5A. There areoverlapping peaks at different concentration in fig. 5A and no overlaping values in fog. 5 B. Besides the peak current values in the calibration line are much smaller than those found in the SWV plot.
2. It is claimed that the value of LOD is 0.30 micromol/l but accroding to the calibration equation at this value the current should be negative.
3. The performance of the various sensing materials used for morin is based only on LOD. In fact, the most important electroanalytical parameter is the concentration range of linear response. This information is missing (see Table 3) and thus it is difficult to make a conclusion on the progress made in this work.
4. The interference study clearly shows that rutin interferes in a marked way with the electroanalytical signal. With increasing morin concentration the peak shifts to more positive potentials and thus the argument on the preservation of the current value at constant potential (0.46 V) is certainly not valid. It is also most confusing that the interference is made at morin concentration beyond the concentration range of linear response found in the present study.
5. The same is valid for the measurements with the chamomile beverage (fig. 7). The additions are far beyond the concentration range of linear SWV response. The plotted straight line in fig. 7B seems not to be the best fit of the four experimental points. The best fitted line would give much larger concentration of morin in the real sample.
Apart from the electroanalytical part it should be also noted that the scan rate dependences (figs. 1 and 3) are studied in narrow scan rate interval. The scan rates should vary within at least one order of magnitude. It is also quite surprising to see the addition of the surface areas obtained from the cathodic and anodic branches of the voltammetric peaks under the name "Total area"(Table 1).
The technical quality of the manuscript should be markedly improved. All plots should have bars at the two axis showing the exact position of the corresponding numbers.
Reviewer 4 Report
The submitted article, "Electrochemical determination of morin in natural food using a chitosan-graphene glassy carbon modified electrode" addresses an important topic related to electrochemical sensing of polyphenol antioxidants. I regret to inform you that the aim of this report is rather than safe as follows:
1. The novelty statement of this study is the use of chitosan-graphene based sensors to determine Morin electrochemically. Several papers have been published in which Morin has been determined, and detection limits have been improved. Listed below are a few of them:
1.1. Electrochemical determination of morin in Kiwi and Strawberry fruit samples using vanadium pentoxide nano-flakes (https://doi.org/10.1016/j.jcis.2017.03.039).
1.2. Hexammine cobalt(III) coordination complex grafted reduced graphene oxide composite for sensitive and selective electrochemical determination of morin in fruit samples (https://doi.org/10.1039/C8QI00055G).
1.3. Electrocatalytic Oxidation and Determination of Morin at a Poly(2,5-dimercapto-1,3,4-thiadiazole) Modified Carbon Fiber Paper Electrode (https://doi.org/10.1149/2.0021609jes).
1.4. Sensitive voltammetric determination of Morin in Psidium guajava leaf extract at Nickel (II) phthalocyanine modified carbon paste electrode (https://doi.org/10.1016/j.surfin.2020.100517).
1.5. Determination of morin on an electrochemically activated carbon-paste electrode https://doi.org/10.3906/kim-1805-37
2. In lines 96-97: Only the freshly modified 96 Ch-G/GCE electrode was treated by applying a potential constant of -1.0 V for 60 s in BPS 97 with a pH of 7.0. There is no mention of the reason for applying a potential of -1.0 V in the study.
3. On the Y-axis, the unit of the measured current should be uniform A or µA.
4. In lines 234 and 235: Net SWV currents were used under optimized conditions: pH of 3.0 (PBS), 0.0 V by 234 30 s in the accumulation step, frequency of 15.0 Hz, and amplitude pulse of 0.05 V in the 235 scan step. There was no mention of how these optimizations are implemented or where the figures illustrating them can be found.
5. In the Interference study and reproducibility section (line 252): Figure 6A illustrates two separate voltammetry graphs for MR and RT, but the effect of interference is only visible if the interfering agents are added to the MR solution. Figures are not provided to illustrate the addition of QC, RT, and CT to the MR solution.
6. MR, QC, RT, and CT are all polyphenolic antioxidants with similar chemical structures. I question why chitosan-graphene is unique in that it can determine MR without interfering with other polyphenols.
According to the comments listed above, the paper should be rejected.